# TRI-TENSE FORMER: CAPTURING DYNAMIC TRAFFIC FLOW USING TRI-TENSE ATTENTION FOR TRAFFIC FORECASTING

## ABSTRACT

Accurate traffic forecasting is essential to enable advanced utilization of intelligent transportation systems. However, forecasting models often struggle to capture the complex spatio-temporal dependencies of traffic data, as they typically handle spatial and temporal dependencies separately. To overcome this limitation, we introduce the Tri-Tense Former (TTformer), a novel approach that captures spatio-temporal relationships through three tense-specific attention modules. We categorize traffic flow into three tense dimensions: past-to-present (present-perfect), present, and future. Each tense-specific attention module captures the dependencies within its respective traffic flow. Furthermore, to address incomplete traffic data, we improve the robustness of the model by employing contrastive learning with negative filtering technique that operates regardless of predefined adjacency matrices. TTformer significantly outperforms existing models by more effectively capturing spatio-temporal dependencies and improving traffic forecasting accuracy.

## 1 INTRODUCTION

Traffic has a significant impact on daily life, influencing the economy, environment, and public safety. Therefore, accurate traffic forecasting is critical for developing more efficient and safer cities. However, predicting traffic flow remains a challenging problem due to the intricate spatial and temporal dependencies within traffic data (Zhao et al., 2019). In response, machine learning research has focused on addressing both the spatial dependency between roads and the temporal dependency, where past traffic states influence current conditions. A common approach is to design separate modules to address each type of dependency (Sahili & Awad, 2023). Recurrent Neural Networks (RNNs), which excel in time series prediction, are typically used to capture temporal characteristics. For spatial features, graph-based models, which treat road networks as graphs, have proven more effective than Convolutional Neural Networks (CNNs). Recently, with the rise of transformer architectures, attention-based models have gained traction as a new trend in this field. Many researchers have thus factorized spatial and temporal attention, aligning with the broader trend in spatio-temporal research. However, to achieve a more comprehensive understanding of spatio-temporal dependencies, it is essential to capture these relationships in their entirety, rather than in isolation (Grigsby et al., 2021). In light of this, we introduce a model that holistically captures spatio-temporal relationships by leveraging attention mechanisms from a novel perspective.

As illustrated in Figure 1a, the vanilla spatio-temporal model employs two aspects of the attention mechanism: spatial and temporal. The spatial attention mechanism captures the relationship between roads at the same timestamp, while the temporal attention mechanism identifies relationships between timestamps for the same variable. However, in this setting, it becomes challenging to quantify the impact of past traffic conditions on the current state, as it doesn't account for how past traffic on other roads influences the present. Therefore, a novel approach to the attention module is needed to directly represent these complex traffic relationships.

To effectively represent the intricate dependencies in traffic conditions, separating them into purely spatial and temporal dimensions is not ideal (Grigsby et al., 2021). Instead, we focus on capturing the real traffic flow to simultaneously represent spatio-temporal relationships. These relationships

can be categorized into three types of traffic flow: from past to present, the present, and from present to future. Using the concept of tense, these flows correspond to the present-perfect, present, and future tenses, respectively.

We propose a model called the "Tri-Tense Former (TTformer)," which incorporates three tense-specific attention modules, each reflecting one of these tense-traffic flows, as shown in Figure 1b. The present-perfect tense attention module quantifies the impact of past traffic conditions on the current traffic state in a spatio-temporal context. The present tense attention module captures the impact of spatial relationships between roads on the current traffic state. Lastly, the future tense attention module measures the prospective temporal relationships, capturing how current traffic conditions influence future states. To implement each tense-attention module, we organize the traffic sequence into "spatio-temporal blocks" and assign an appropriate pair of query and key vectors in the attention mechanism. Additionally, traffic data often contains missing values or outliers, primarily due to sensor failures or inaccuracies (Liu et al., 2022). To address these imperfections, we adopt a contrastive learning method to derive more robust representations. Our approach also includes a negative filtering methodology, which proves especially useful in the absence of a predefined adjacency matrix.

The main contributions of this paper are as follows:

- We propose Tri-Tense Former (TTformer), a model that captures spatio-temporal relationships with three tense attention modules: the present-perfect, the present, and the future, reflecting real traffic flow dynamics. We implement this by structuring traffic sequences into 'spatio-temporal blocks' and setting three different key-query pairs in the attention mechanism.

- We adopt contrastive learning to address inconsistencies in traffic datasets. In the negative filtering step of contrastive learning, we use spatial embedding instead of a predefined adjacency matrix, enabling the methodology to be effective even when predefined information is unavailable.

- We conduct a series of experiments using real traffic datasets, METR-LA and PEMS-BAY. The proposed model demonstrates superior performance compared to baseline models, indicating that TTformer is capable of jointly capturing spatio-temporal relationships effectively.

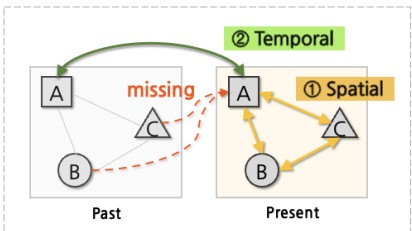 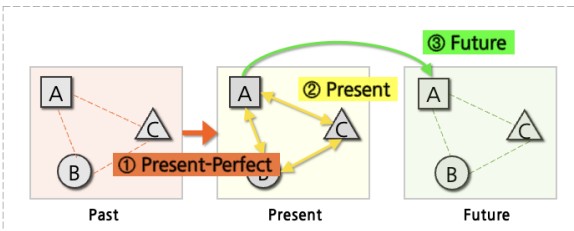

(a) Vanilla Spatio-temporal Attention         (b) Tri-Tense Attention

Figure 1: Spatio-Temporal Attention mechanism

## 2 RELATED WORK

### 2.1 SPATIO-TEMPORAL TRANSFORMER

The attention mechanism is capable of capturing dynamic relationships, and it has no limit on the receptive field, allowing it to handle long dependencies. Consequently, Transformer-based models are currently being researched to identify relationships between roads using attention mechanisms instead of adjacency matrices (Dwivedi & Bresson, 2020). The attention score $Att(L_q, L_k)$ calculated between a query sequence $L_q$ and a key sequence $L_k$ represents the relationship between the query and the key. This allows the model to dynamically learn and adapt to the relationships between roads based on the traffic context, rather than relying on predefined static adjacency matrices.

Traffic forecasting models that implement attention mechanisms compute both spatial and temporal attention to perform prediction tasks. Numerous models such as Dsformer (Yu et al., 2023) and Traffic transformers(Cai et al., 2020) have been developed to calculate spatial and temporal attention separately.

When calculating attention separately for spatial and temporal dimensions, a limitation exists in that only temporal relationships per sequence are expressed. Therefore, models that capture spatio-temporal relationships simultaneously are being studied, allowing for a more comprehensive understanding of the interactions between variables across both spatial and temporal dimensions. Space-timeformer (Grigsby et al., 2021) tries to address this issue by reorganizing tokens to express spatio-temporal attention. The model handles $N$ nodes and $(c+h)$ time steps simultaneously, thus enabling the calculation of spatio-temporal relationships in a unified manner. Spacetimeformer trains a global self-attention network that jointly computes attention across both space and time for simultaneous comprehension of spatio-temporal relationships. However, this approach leads to a quadratic increase with $(N * (c + h))$ in computational complexity and memory requirements. Therefore, additional techniques are required to reduce the computational complexity by sparsifying the computational matrix (Zhou et al., 2021). STAEformer (Liu et al., 2023) employs an adaptive embedding mechanism to capture spatio-temporal relationships simultaneously. However, the adaptive embedding has the disadvantage of maintaining a static representation, which may not adequately represent the dynamic spatio-temporal relationships that may change according to the traffic state.

To capture the spatial and temporal dependency, numerous GNN-based models and the attention-based models factorize each dependency for calculation. However, this approach results in insufficient capture of spatio-temporal dependencies. Consequently, we present a novel approach to capture hidden dependencies by introducing three tense attention modules that can simultaneously understand spatio-temporal relationships. When compared to Spacetimeformer, our model offers significant scalability advantages in that it maintains a constant graph size, which is the same as when computing spatial and temporal attention separately.

## 2.2 Traffic Data Imputation

Addressing the incompleteness of traffic datasets is crucial for enhancing the performance of traffic prediction models. Events such as unexpected accidents, broken sensors, or missing signals frequently occur, significantly decreasing the quality of the collected data (Fang & Wang, 2020). Simple deletion of missing values is only feasible when the missing rate is low (Wothke, 2000; McKnight et al., 2007). Therefore, various data imputation methods have been developed. Initially, neighbor-based methods provided a solution by identifying the nearest neighbors using techniques like KNN and updating the missing values with the average of these neighbors (Song et al., 2015) or replacing with the last observed value (Amiri & Jensen, 2016). Recently, deep learning-based methods have demonstrated superior performance, encompassing conventional learning techniques—both supervised and unsupervised learning—and more recent methods such as self-supervised learning.

RNN-based models such as GRU-D (Che et al., 2018), BRITS (Cao et al., 2018), and NAOMI (Liu et al., 2019) are effective for handling time-series datasets. While they have proven effective in handling missing data within time series datasets, they are not specifically designed for spatio-temporal datasets. These models primarily focus on temporal dependencies and do not adequately account for spatial relationships among adjacent nodes in networks. These limitations underscore the need for novel approaches that can integrate both spatial and temporal dependencies, leveraging the relational information inherent in spatio-temporal datasets. GRIN (Cini et al., 2022) tackles the shortcomings of conventional time-series imputation techniques by employing a graph structure to effectively handle spatio-temporal data and learning representations through message passing with RNN cell. Like the BRITS model, GRIN processes multivariate time series data in both forward and backward directions at each node, operating bidirectionally. However, GRIN is prone to the error propagation issues typically found in autoregressive models. SPIN (Marisca et al., 2022) was developed to mitigate the error propagation issue observed in GRIN. SPIN leverages attention-based architectures instead of relying on recurrent neural networks to avoid the autoregressive propagation of errors. SPIN employs inter-node spatio-temporal cross-attention to propagate information from neighboring nodes and intra-node temporal self-attention to analyze the sequence of each node independently. STGAIN (Huang et al., 2023) represents another advanced approach, utilizing GANs

for robust spatio-temporal imputation. It features a spatio-temporal generator and a discriminator that incorporates a GCN as a spatial aggregator and a 1-D CNN as a temporal extractor.

While conventional learning techniques have demonstrated good performance, they require dedicated training for a data imputation model. Contrastive learning offers a potential solution to this limitation by not directly imputing the missing values but enhancing the quality of the learned representations for the prediction. The objective of contrastive learning is to increase the similarity between positive pairs while decreasing it for negative pairs, thereby refining the representations to capture meaningful information. STGCL (Liu et al., 2022) highlights data scarcity as an obstacle to the performance of spatio-temporal forecasting and argues that the application of contrastive learning is beneficial. The author proposes several data augmentation methods and negative filtering techniques. With regard to spatial negative filtering, the authors utilize the known adjacency matrix to classify first-order neighbors as hard negatives and exclude them from contrastive loss computation. However, this approach is subject to the limitation of requiring a predefined matrix. Therefore, we suggest an alternative method that can be applied without the need for a predefined matrix.

## 3 METHODOLOGY

**Preliminary.** Given a set of $c$ timestamps of $N$ variables, the objective is to predict the next $h$ timestamps. To formulate the problem, we adopt the graph concept to model traffic conditions, where each variable represents sensors on the road with traffic attribute values. Each graph, designated as $G_t = (V_t, A_t)$, represents the traffic state at the given timestamp $t$. Consequently, as illustrated in Figure 2, the traffic forecasting problem can be formulated as follows: when the context sequence of graphs $G[C] = [G_{T-c+1}, \cdots, G_T]$ is given, the target sequence of graphs $G[H] = [G_{T+1}, \cdots, G_{T+h}]$ is to be predicted. Each node in the graph, designated as $v_t^i$, represents the $i^{th}$ sensor on the road. For $N$ traffic sensors, the total number of nodes in the graph is equal to $|V_t| = N$. The adjacency matrix, $A_t$, is a matrix of $A_t \in \mathbb{R}^{N*N}$ that represents the relationship between $N$ roads. In the context of traffic forecasting, a predefined static adjacency matrix, calculated based on the distances between roads, is typically employed as auxiliary information. In contrast, this paper presents a model that predicts traffic flow without relying on predefined relationships between roads.

Additionally, in traffic forecasting, since traffic data includes periodicity according to timestamps (such as time of the day and day of the week), it is common practice to use both traffic attributes $x$ and timestamp information $y$ together (Zhou et al., 2021). In the graph $G_t = (V_t, A_t)$, each node represents the $i^{th}$ variable at timestamp $t$, containing timestamp information $x_t^i$ and traffic attribute values $y_t^i$. This node is represented by the vector $v_t^i = [x_t, y_t] \in \mathbb{R}^d$, where $d$ is the dimension of each node's value. In this scenario, accurate traffic forecasting relies on the representation of each node $v_t^i$, ensuring that each representation effectively captures the spatio-temporal dependencies. Spatial dependency refers to the spatial relationships between variables, while temporal dependency refers to the relationships over time for a single variable along the time axis.

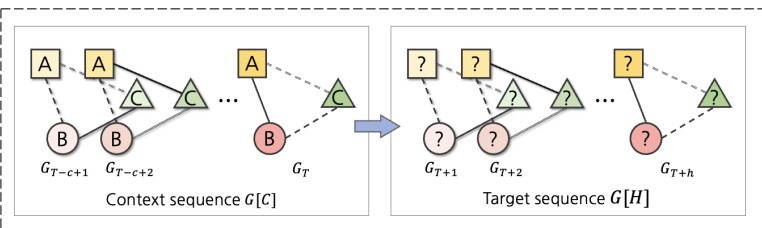

Figure 2: Traffic forecasting problem

### 3.1 SPATIO-TEMPORAL BLOCK

When dealing with long context and target sequences, transformer-based models face limitations in computational complexity and memory consumption. To address this challenge, we introduce a novel concept called the "spatio-temporal block". This approach improves the capability of trans-

former models to effectively handle long-range prediction tasks. Please refer to Appendix A.1 for details.

## 3.2 Tri-Tense Attention Module

The proposed model is based on the Graph Attention Network(GAT) (Velickovic et al., 2017), which is derived from the Graph Neural Network(GNN) (Scarselli et al., 2008). The proposed model, Tri-Tense Former (TTformer), integrates three attention modules to capture dependencies across both space and time, following the GAT framework. Each module assesses the varying relative importance of each node from three different perspectives, enabling the model to effectively capture spatio-temporal dependencies. Figure 3 illustrates the architecture of TTformer, which follows the encoder-decoder based transformer structure. The overall flow of TTformer can be summarized as follows: both the context and target sequences are updated within the encoder, with the target sequence further refined in relation to the context sequence within the decoder. The contextualized representations of the context and target sequences are then concatenated, and final predictions for the target sequence are generated. The encoder employs two attention modules: the present-perfect attention and the present attention module, while the decoder utilizes a single attention module, the future attention module. Detailed descriptions for the encoder and decoder can be found in Appendix A.3 and A.4.

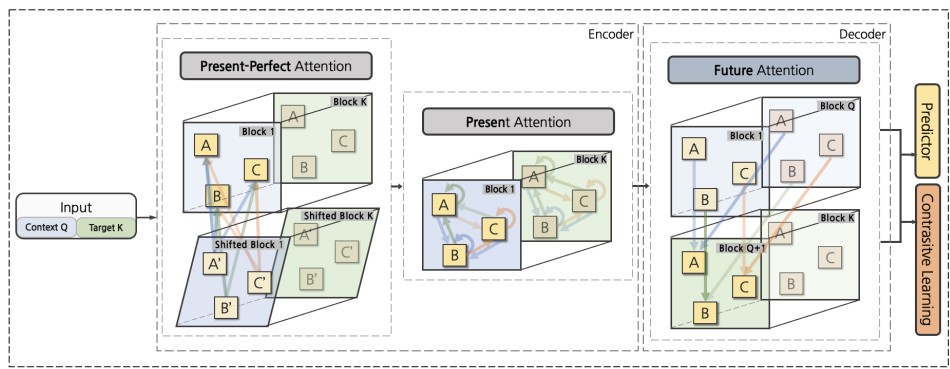

Figure 3: Tri-Tense Former(TTformer)

## 3.3 Contrastive Learning

The framework of contrastive learning is illustrated in Figure 4. Contrastive learning is one of the self-supervised learning techniques that aim to learn effective representations by comparing and contrasting between input data and augmented input data (see Appendix A.5 for data augmentation). Through contrastive learning, the representations of positive pairs tend to become more similar, while those of negative pairs become less similar. This enables the identification of precise representations for input data.

Moreover, the process of negative filtering can be applied further by filtering out hard negatives (see Appendix A.6) for details). These are instances that are closely aligned with the positive pairs but designated as negative pairs. In the cited paper (Liu et al., 2022), the authors proved that contrastive learning is effective in overcoming challenges in traffic forecasting. Regarding the spatial negative filtering method, they proposed utilizing a predefined adjacency matrix to filter out first-order neighbors as hard negatives, thus excluding them from the computation of the contrastive loss. However, in this study, we assume that a predefined adjacency matrix is unavailable. Instead, we propose an alternative negative filtering method that generates a similarity matrix using spatial embeddings.

We adopt a joint learning approach, where both the prediction task and the contrastive task are executed simultaneously. The overall loss is calculated as the sum of the prediction loss, $L_{\text{pred}}$, and the contrastive task loss, $L_{\text{contrastive task}}$ (1). The impact of the contrastive task can be controlled by adjusting the value of the parameter $\lambda$.

$$L = L_{\text{pred}} + \lambda L_{\text{contrastive task}} \tag{1}$$

The contrastive loss suggested by GraphCL (You et al., 2020) is employed, which is a common approach in graph contrastive learning. For the representation $z$ and the augmented representation $z'$, pairs from the same nodes $(z_i, z'_i)$ are designated as positive pairs, while pairs with different nodes $z_i, z'_j$ $(i \neq j)$ are classified as negative pairs. For a set of $M$ nodes, each node is assigned $M-1$ negatives. As the contrastive loss (2) decreases, the refined representations of positive pairs become more similar, while the negative pairs' become less similar.

$$L_{\text{contrastive task}} = \frac{1}{M} \sum_{i=1}^{M} -\log \frac{\exp(\text{sim}(z_i, z'_i)/\tau)}{\sum_{j=1, j\neq i}^{M} \exp(\text{sim}(z_i, z'_j))/\tau)} \tag{2}$$

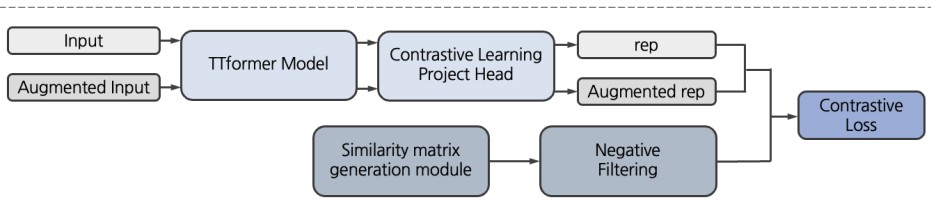

Figure 4: Framework of Contrastive Learning

## 4 EXPERIMENTS

**Datasets.** In the experiments, the real traffic datasets, METR-LA and PEMS-BAY, are used as described at Appendix B.1. The objective of the traffic forecasting task is to predict 12 timestamps (1 hour) of the target sequence for the given 12 timestamps of the context sequence. Following the traffic forecasting baseline models, the data was normalized using a z-score.

### 4.1 COMPARISON WITH BASELINE MODELS

**Baselines.** To highlight the efficacy of TTformer, we select five spatio-temporal traffic forecasting models as baselines. The baselines are LSTM (Hochreiter & Schmidhuber, 1997), DCRNN (Li et al., 2018), GraphWavenet (Wu et al., 2019), STGCN (Han et al., 2020), and STAEformer (Liu et al., 2023). Figure 5 presents the prediction results of the proposed TTformer and all baseline models across two datasets, with additional details available in Table 1. The best results are highlighted in bold, while the second-best results are underlined. Here, the performance of STAEformer* is as reported in the paper by (Liu et al., 2023). When compared to the plain LSTM, the Spatio-Temporal GNN models (Li et al., 2018; Wu et al., 2019; Han et al., 2020) exhibited a significant enhancement in prediction performance. Among these GNN-based models, DCRNN showed the overall most promising performance across both datasets. Moreover, the attention-based model, STAEformer, demonstrated superior performance compared to GNN-based models, underscoring the importance of capturing dynamic spatial relationships between roads for predictive accuracy. A direct comparison between STAEformer and TTformer revealed that TTformer surpasses STAEformer, proving the effectiveness of the proposed architecture in capturing spatio-temporal relationships with three tense aspects. The label 'TTformer(w/o CL)' denotes the TTformer without the contrastive learning technique. Despite not employing contrastive learning, TTformer still delivers commendable performance, indicative of the robustness of its three tense-attention modules design.

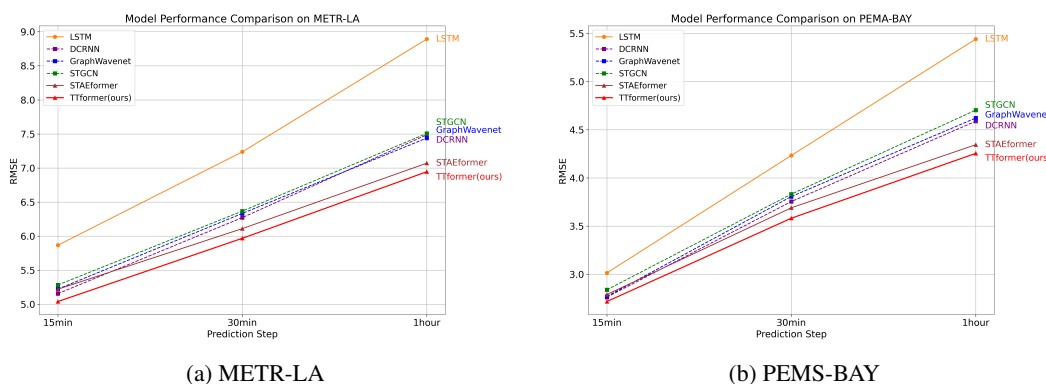

(a) METR-LA             (b) PEMS-BAY

Figure 5: Performance comparison of different approaches(RMSE)

Table 1: Comparison of different approaches for traffic forecasting

| | T | Metric | LSTM | DCRNN | GraphWavenet | STGCN | STAEformer* | *TTformer* | *TTformer(w/o CL)* |
|---|---|---|---|---|---|---|---|---|---|
| **METR-LA** | 15min | *RMSE* | 5.87 | 5.16 | 5.22 | 5.28 | 5.11 | **5.04** | 5.10 |
| | | *MAE* | 2.98 | 2.66 | 2.71 | 2.75 | 2.65 | **2.63** | 2.66 |
| | | *MAPE* | 7.96 | 6.84 | 7.11 | 7.00 | 6.85 | **6.79** | 6.92 |
| | 30min | *RMSE* | 7.24 | 6.27 | 6.33 | 6.37 | 6.00 | **5.97** | 6.02 |
| | | *MAE* | 3.58 | 3.07 | 3.11 | 3.16 | 2.97 | **2.95** | 2.97 |
| | | *MAPE* | 10.17 | 8.36 | 8.70 | 8.39 | 8.13 | **8.07** | 8.18 |
| | 60min | *RMSE* | 8.89 | 7.49 | 7.44 | 7.51 | 7.02 | **6.95** | 7.02 |
| | | *MAE* | 4.42 | 3.55 | 3.56 | 3.64 | 3.34 | **3.31** | 3.33 |
| | | *MAPE* | 13.61 | 10.27 | 10.38 | 10.04 | 9.70 | **9.58** | 9.65 |
| **PEMS-BAY** | 15min | *RMSE* | 3.01 | 2.76 | 2.77 | 2.84 | 2.78 | **2.72** | 2.77 |
| | | *MAE* | 1.40 | 1.31 | 1.31 | 1.36 | 1.31 | **1.30** | 1.317 |
| | | *MAPE* | 2.94 | **2.71** | 2.77 | 2.86 | 2.76 | 2.73 | 2.77 |
| | 30min | *RMSE* | 4.23 | 3.76 | 3.81 | 3.83 | 3.68 | **3.58** | 3.69 |
| | | *MAE* | 1.85 | 1.65 | 1.66 | 1.71 | 1.62 | **1.60** | 1.63 |
| | | *MAPE* | 4.21 | 3.65 | 3.73 | 3.84 | 3.62 | **3.60** | 3.69 |
| | 60min | *RMSE* | 5.44 | 4.59 | 4.62 | 4.70 | 4.34 | **4.26** | 4.35 |
| | | *MAE* | 2.37 | 1.97 | 1.98 | 2.06 | 1.88 | **1.85** | 1.89 |
| | | *MAPE* | 5.90 | 4.58 | 4.68 | 4.82 | 4.41 | **4.31** | 4.47 |

The performance results for STAEformer* are taken from the author's paper Liu et al. (2023), while the results for the other baseline models are obtained from experiments conducted under the settings described in Appendix B.2.

## 4.2 ANALYSIS OF TTFORMER ATTENTION MODULE

### 4.2.1 ABLATION STUDY OF THREE TENSE-ATTENTION MODULES

The subsequent step is to examine the tense-attention modules of TTformer. An ablation study was conducted by training the model while excluding each of the three modules. The results of the performance comparison on the METR-LA test dataset when contrastive learning was jointly trained are shown in Figure 6a. In Figure 6a, 'w/o 1st' signifies the model that excludes the first tense-attention module, 'w/o 2nd' means the second, and 'w/o 3rd' means excluding the third module. And the last 'w/ random 3rd' means for replacing the third future attention module with a random future attention module.

The 'TTF' case represents the scenario where all three attention modules are applied, resulting in the best performance. Among 'w/o 1st', 'w/o 2nd', and 'w/o 3rd', the best-performing model is 'w/o 3rd', which excludes the third module. In this instance, it can be inferred that both the first and second modules effectively reflect the relationships for edge types A and B as described in Figure 10, and capture the spatio-temporal relationships well. Among the three altered models, the exclusion of the second module results in the largest performance drop. This indicates that the current attention mechanism has the greatest influence on the predictions.

The third module is responsible for capturing future attention in the context sequence, thereby updating the target sequence using the calculated future attention. The removal of the third future attention module resulted in a decline in performance. This suggests that the third module plays a role in up-

dating the target sequence. Additionally, in order to confirm whether the future attention learned was indeed effective, a further study was conducted to replace the model's third future attention module with a randomly generated future attention module. In the proposed TTformer model's architecture, the first and second modules are maintained in their original form; however, as shown in Figure 6b, the last future attention module is replaced with the random future attention module. This allows the target sequence to randomly refer to the input sequence with random attention scores. The random attention scores are generated by applying the softmax function to random values, which are then used for a weighted sum on the input sequence. The comparison results demonstrated that the Test MAE of the random future attention module is 2.926, while the Test RMSE is 5.954. In contrast, the Test MAE of the model with learned future attention('TTF') is 2.912, and the Test RMSE is 5.911. This confirms that the incorporation of trained future attention contributes to an enhancement in the model's performance.

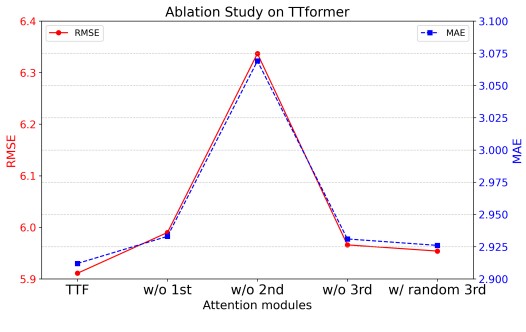
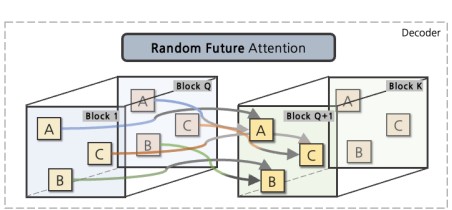

(a) Ablation study on METR-LA       (b) Random future attention module

Figure 6: Ablation study of TTformer tense-attention module

### 4.2.2 QUALITATIVE ANALYSIS OF 2 TENSE-SPECIFIC ATTENTION MODULES IN ENCODER

Next, we conducted a qualitative analysis between the present-perfect attention and present attention modules, using a case study centered around Dodger Stadium. For this analysis, we've selected 10 roads around the Dodger Stadium to compare the efficacy of these two tense-attention modules.

Reverting back to June 13, 2012, at 19:00 on a Wednesday evening, Dodger Stadium in Los Angeles, California, hosted a significant baseball match between the Los Angeles Angels of Anaheim and the Los Angeles Dodgers, drawing an attendance of 43,494. This notable event provides a context for examining traffic dynamics in the vicinity. Notably, we observed a speed drop on road 93 around the time of the game, as depicted in Figure 7b.

In Figure 8, we visualize two tense-attention matrices among selected 10 roads focusing on the time-series data from 18:40 to 19:00, which is right before the game starts. Firstly, in Figure 8a, we illustrate the present-perfect attention matrix, which captures the influence of past states on current states. Noteworthy is the discernible impact of prime roads, such as road 144 and road 91 leading toward Dodger Stadium. Conversely, roads 93, 6, and 23, characterized by relatively lower attention values, represent outgoing routes from the stadium. This suggests that the past states of roads leading toward Dodger Stadium exerted a considerable influence on present road conditions.

Contrastingly, the attention matrix from the present attention module as depicted in Figure 8b, focuses primarily on current flow states, showcasing a different phenomenon. Here, attention is concentrated more on the road itself and its adjacent counterparts. This observation highlights the distinct roles of the two attention modules, aligning well with the intuitive expectations.

Through this comparative visualization, we have confirmed the divergent roles of the two tense-attention modules. The present-perfect attention effectively captures historical influences on the current state, while the present attention module emphasizes current dependencies. These insights offer valuable perspectives for understanding traffic dynamics near Dodger Stadium.

Furthermore, we analyze the spatio-temporal blocks in Appendix B.3 and contrastive learning in Appendix B.4.

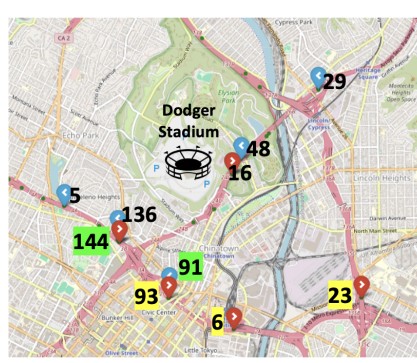
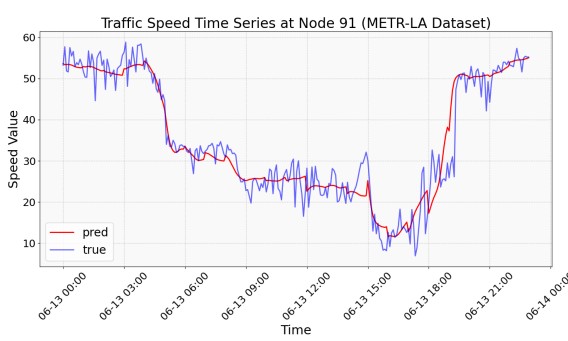

(a) Roads near the Dodger Stadium

(b) Time-series on the road 91

Figure 7: Traffic Analysis on Dodger Stadium Game Day

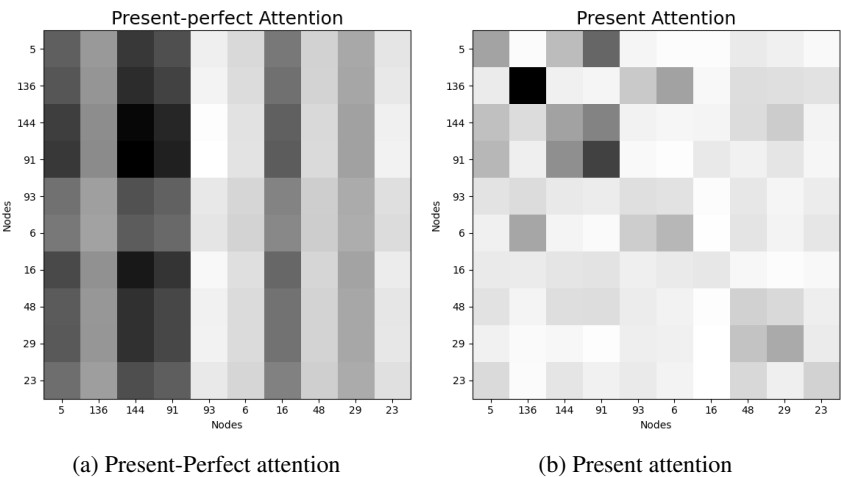

(a) Present-Perfect attention

(b) Present attention

Figure 8: Comparative analysis of the Present-Perfect Attention and Present Attention modules

The figure shows two attention matrices from each module, focusing on the time-series data from 18:40 to 19:00 on June 13, 2012, during a significant event at Dodger Stadium.

## 5  CONCLUSION

In contrast to conventional approaches that factorize traffic dependency into spatial and temporal aspects, we introduce a novel perspective of attention modules to express the three types of traffic flow -present-perfect, present, and future- through each corresponding tense-specific attention module. By capturing each tense relationship using a respective attention module, the proposed model, Tri-Tense Former(TTformer), excels in traffic forecasting tasks. Furthermore, to address the unstable characteristics of traffic datasets, which may contain high ranges of zero values, missing values, or outliers, we integrate contrastive learning techniques with an innovative negative filtering method.

We demonstrate the efficacy of the proposed model with two real traffic datasets: METR-LA and PEMS-BAY. In comparison with baseline models, TTformer outperforms other baseline models by capturing hidden dependencies within the data. Moreover, comprehensive ablation studies validate the necessity of each proposed module for capturing intricate spatio-temporal relationships. In the context of contrastive learning, our findings align with prior research (Liu et al., 2022), highlighting the efficacy of node masking as a data augmentation strategy. Additionally, the proposed negative filtering method, which utilizes spatial embedding to generate similarity matrices, demonstrates superior predictive performance compared to conventional approaches using predefined adjacency matrices.

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

## A   TRI-TENSE FORMER

### A.1   SPATIO-TEMPORAL BLOCK

The spatio-temporal block is defined as the merging of a graph for a specific time interval, referred to as the "block size" $S$. Figure 9 illustrates the methodology employed in the design of the spatio-temporal block when the block size $S = 3$. By merging $S$ size of graphs, each block contains all the information pertaining to the total variables within the specified block size. Each node is now redesigned to contain information for the time interval $(t, t+1, ..., t+S-1)$ rather than a specific time $t$. The initial block is represented as $B_1 = [G_{T-c+1}, G_{T-c+2}, ..., G_{T-c+S}]$, with the node within $B_1$ being $V_{B_1} = [V_{T-c+1}, V_{T-c+2}, ..., V_{T-c+S}]$. The integration of the spatio-temporal block leads to a reduction in the total graph size from $N * (C + H)$ to $\frac{N*((C+H))}{S}$.

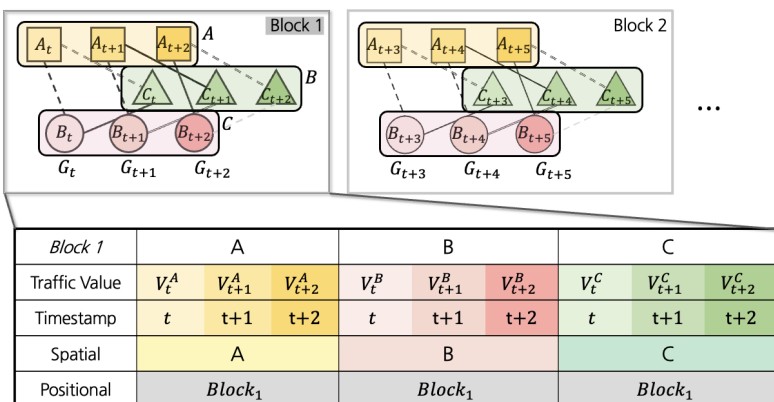

Figure 9: Spatio-temporal block

The utilization of the spatio-temporal block differs significantly from the application of dilated causal convolution layers in WaveNet (Oord et al., 2016). In WaveNet, dilated convolution is employed to train a kernel, facilitating the extraction of temporal trends. This allows for the handling of relatively large receptive fields with few layers and enables the parallel processing of sequences, contrasting with recurrent neural networks. However, the spatio-temporal block operates in a different manner. Unlike dilated convolution in WaveNet, its function is not to extract temporal trends through training the kernel. Instead, it simply merges nodes along the time axis. This enables the aggregation of information across multiple time steps within a specified block size, facilitating the modeling of spatio-temporal relationships. The spatio-temporal block offers three principal advantages. Firstly, the complexity of the training is reduced as the size of the total graphs is decreased. Secondly, it enhances the effectiveness of transformer-based models by facilitating more efficient application of the attention mechanism. When applying an attention-based model to a large graph, additional techniques such as sparsifying the attention matrix are required. However, the spatio-temporal block simplifies this process by leveraging a simple inductive bias that traffic attributes exhibit temporal dependency within a specific temporal range. This can reduce the size of the attention matrix itself, eliminating the need for additional techniques for the attention mechanism. Thirdly, the application of spatial attention within the spatio-temporal block inherently incorporates the effect of temporal attention. This is because each block contains traffic flow within a specific block size, allowing temporal relationships to be considered by calculating the spatial attention operation within the same block. The experimental results, which vary according to the block size, will be presented in the subsequent section.

## A.2 NODE EMBEDDING

The process of embedding each node, as depicted in Figure 9, involves incorporating various types of data: traffic attributes, temporal information, spatial information for distinguishing each variable, and positional information indicating the sequence of the block. In the context of traffic forecasting, the traffic attribute value can be represented by either traffic speed or traffic volume. Embedding the traffic value involves applying a single linear layer projection. Temporal information is represented using a lookup table consisting of two temporal variables: the timestamp of the day and the day of the week. The final two embedding, spatial embedding and positional embedding, are set as learnable parameters.

In a transformer-based model, maintaining awareness of the order of elements within a sequence is essential. Unlike RNN-based models, which process sequences sequentially, transformer models ingest the entire sequence simultaneously. Therefore, to clarify each node $v^i_{Block\_s}$, spatial embedding is employed to convey the position of each element ($i^{th}$ variable) and positional encoding assists in recognizing positions of the block ($Block\_s$).

In the case of the target sequence, the traffic attribute remains unknown and requires prediction. Hence, during training, the traffic value of the target sequence is initialized as a zero vector (Zhou et al., 2021).

## A.3 ENCODER : PRESENT-PERFECT ATTENTION & PRESENT ATTENTION MODULE

The objective of the encoder is to capture the dynamic relationships between nodes within each block. As traffic flow fluctuates, so do the relationships between variables. In the proposed model, both the present-perfect attention module and the present attention module are employed to capture these dynamic relationships. The need for two different types of attention modules arises from the following rationale: to fully capture the relationships between nodes, it's essential to consider not only the current state but also the past state, as recent past states also play a significant role in shaping the current relationships. The visualization of these influences is illustrated in Figure 10, where nodes at past timestamps also impact current nodes. In order to understand the spatial relationship between $A_{t+1}$ and $B_{t+1}$ and $C_{t+1}$, the past state $A_t$, $B_t$, and $C_t$ must also be taken into account. When viewed in the context of a single large graph, two types of edges can be identified: edge type A represents the influence of the past on the current state, while edge type B represents the influence of the current state on the current state. This concept can be elucidated using heterogeneous graphs, where various types of edges are present in a single graph (Hu et al., 2020). In TTformer, each tense-attention module effectively manages these two edge types of temporal edge relationships, addressing the distinct influences of past and present states on the current state.

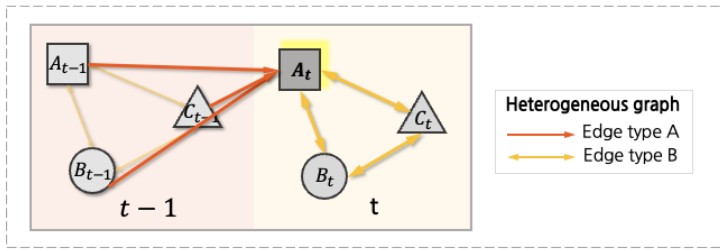

Figure 10: Traffic flow in heterogeneous graph

The first attention module, termed the 'present-perfect attention module,' is designed to quantify the impact of past traffic states on the current spatial relationships between nodes. This module utilizes 'shifted' spatio-temporal blocks, constructed from the given context sequence depicted in Figure 11. These shifted blocks are created by duplicating the first timestamp, shifting the entire sequence one timestamp ahead, and then dividing them into shifted spatio-temporal blocks. Shifting the timestamps allows us to capture the influence of past traffic conditions on the current state. It's reasonable to shift by one timestamp, as the most recent traffic conditions typically exert the greatest influence on the current traffic. To implement this first module, we map the original (spatio-temporal) 'blocks' into a Query vector, while the 'shifted blocks' are mapped into Key and Value vectors.

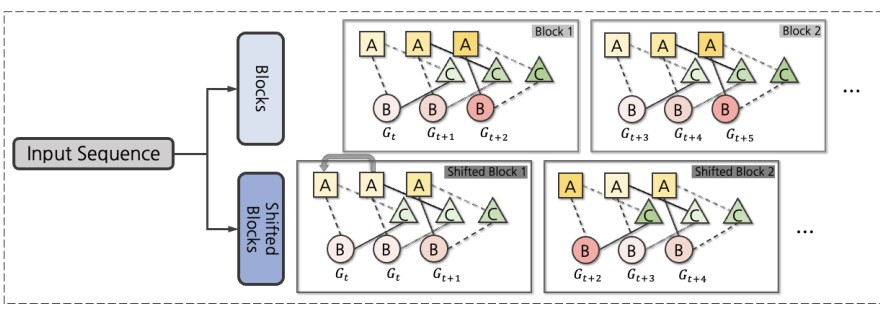

Figure 11: Spatio-temporal shifted blocks

The subsequent module is the 'present attention module'. This module is responsible for calculating the spatial attention based on the current traffic state. To achieve this, each Query, Key, and Value vector utilizes the same 'blocks', constituting a self-attention step.

Both the preceding modules are applied to both the context sequence and the target sequence. For the target sequence, the traffic attribute, temporal information, and positional information are shared within the same block, with only the spatial embedding differing between nodes. Despite sharing many attributes, the application of two attention modules still yielded benefits in updating the representations of nodes in the target sequence. This is attributed to the modules' ability to capture internal relationships between variables that persist regardless of the traffic state.

## A.4 DECODER : FUTURE ATTENTION

The subsequent stage is the decoder, which incorporates the module called the 'future attention module'. This module is employed to capture the temporal relationships between the context sequence, which represents the present, and the target sequence, which represents the future. By computing these temporal relationships between the present and the future, the module updates the representations of the target sequence, discerning meaningful blocks from the context sequence. In this process, the context sequence is utilized as Key and Value vectors, while the target sequence serves as the Query vector. To generate the final output, the output from both the context sequence blocks and the target sequence blocks are concatenated. Following the approach advocated by Informer (Zhou et al., 2021), a generative-style decoder is employed to predict the target sequence in a single step, thereby mitigating the potential accumulation of errors associated with auto-regressive models. The prediction loss, $L_{\text{pred}}$, is computed as the mean absolute error(MAE) between the actual target values, $Y$, and the predicted values, $\hat{Y}$ (3).

$$L_{\text{pred}} = \left| \hat{Y}^{T+1:(T+h)} - Y^{T+1:(T+h)} \right| \tag{3}$$

## A.5 DATA AUGMENTATION

In order to ensure smooth training, it is crucial to perform data augmentation correctly. Among the four methods proposed by STGCL (Liu et al., 2022) for data augmentation, we adopt the node masking technique. They compared contrastive learning with different data augmentation methods using real traffic datasets, PEMS-04 and PEMS08, and concluded that node masking was the most effective. As illustrated in Figure 12, the node masking method involves masking a certain percentage of randomly selected points to zero. It was found that masking only the traffic attribute at a 1% ratio was the most effective. Even if some data within the sequence includes zeros, the agreement is maximized so that the sequence of the same node without zeros can have similar representations in the latent space. This is a useful technique in traffic datasets with a high portion of zero values, as it ensures that predictions remain robust and stable, even when there are zero values in the data.

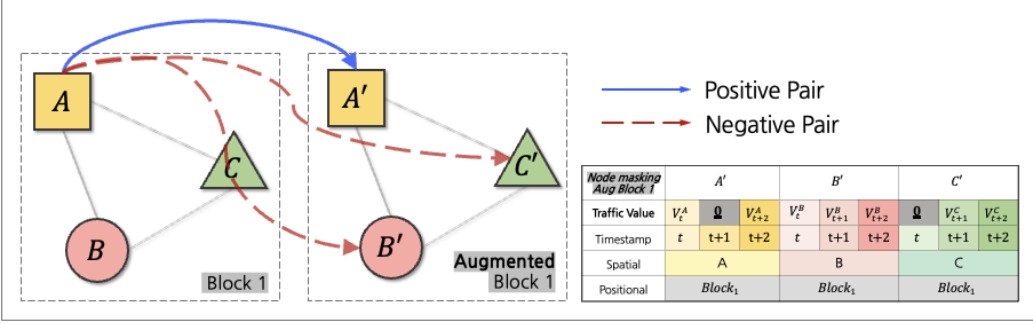

Figure 12: Node masking in contrastive learning

Table 2: Traffic datasets

| Datasets | #Nodes | #Edges | Time span | Interval | Missing Value |
|----------|--------|--------|-----------|----------|---------------|
| METR-LA  | 207    | 1,722  | 34,272    | 5 min    | 8.11%         |
| PEMS-BAY | 325    | 2,694  | 52,116    | 5min     | 0.003%        |

## A.6 NEGATIVE FILTERING

The hard negatives are negative points that are difficult to distinguish from an anchor point(Robinson et al., 2021). Therefore, by filtering out hard negatives, more precise contrastive learning can be achieved. This paper introduces a new negative filtering method to utilize spatial embedding, which are set of learnable parameters and updated during training. As the training progresses, the spatial embedding is trained to represent each variable. Accordingly, the generation of a similarity matrix through the application of a dot product between spatial embedding allows for the acquisition of a similarity matrix. If the similarity between spatial embedding $n_i, n_j \in \mathbb{R}^{node\_emb\_dim}$ is greater than or equal to the filter_threshold, then $node_j$ is classified as a hard negative and excluded from the computation of the contrastive loss (4).

The hard negatives are the negative pairs that are most closely related to the positive pairs. Therefore, by filtering out hard negatives, more precise contrastive learning can be achieved. In this paper, we propose a new negative filtering technique utilizing spatial embeddings, which are sets of learnable parameters updated during training. As the training progresses, the spatial embedding is trained to represent each variable. Accordingly, the application of a dot product between spatial embeddings allows for the acquisition of a similarity matrix. If the similarity between spatial embeddings $n_i, n_j \in \mathbb{R}^{node\_emb\_dim}$ is greater than or equal to the filter_threshold, then $node_j$ is classified as a hard negative and excluded from the computation of the contrastive loss (4).

$$\text{Filter Adj}(i,j) = \begin{cases} 1, & \text{if } i = j \ \lor \ \text{sim}(n_i, n_j) < \text{filter threshold} \\ 0, & \text{otherwise} \end{cases} \quad (4)$$

## B EXPERIMENTS

### B.1 DATASET DETAILS

METR-LA and PEMS-BAY are shown in Table 2. The METR-LA dataset comprises traffic speed data collected from 207 loop detectors in Los Angeles city from March to June 2012. The PEMS-BAY dataset comprises 325 sensors over a six-month period from January to May in 2017.

The characteristic of the two traffic datasets is the high range of zero values. In two datasets, METR-LA and PEMS-BAY, a significant number of zero values are included. The datasets are split into training, validation, and testing, with each portion comprising 70%, 10%, and 20%, respectively. The percentages of zero values in each set are notably high, at 7.297%, 5.733%, and 9.508%, respectively. Similarly, the PEMS-BAY dataset, which is divided into the same proportions as METR-LA, exhibits zero value percentages of 4.855%, 5.647%, and 4.724%, respectively. Furthermore, the PEMS-BAY and METR-LA datasets initially contain missing values, with the percentages of missing values being 0.003% and 8.11%, respectively.

### B.2 HYPERPARAMETERS

**Setting.** The main hyperparameter values of the TTformer are presented in Table 3. And the aforementioned baselines were evaluated using the shared official codes.

### B.3 ANALYSIS OF SPATIO-TEMPORAL BLOCK

The first step of TTformer involves constructing 'spatio-temporal blocks' by aggregating traffic flows over the input sequence. As discussed earlier, the introduction of the spatio-temporal block can

Table 3: Experiment configuration

| Config | Values |
| --- | --- |
| optimizer | Adam |
| learning rate | 0.001 |
| learning rate schedule | ReduceLROnPlateau |
| Dropout | 0.1 |
| episode | 50 |
| batch size | 16 |
| weight decay | 0.0003 |

reduce computational complexity by decreasing the number of nodes and can capture more extensive contextual information reflecting spatio-temporal relationships. However, aggregating data over larger blocks may result in the loss of finer details by not capturing traffic patterns unique to smaller time intervals. Furthermore, as the block size increases, the difference between a key vector from the present-perfect attention and the present perfect decreases, making it difficult to capture different perspectives of relationships through the two discrete attention modules. Therefore, it is crucial to identify the optimal block size.

We compared the model performance while varying the block size $S = [1, 2, 4, 6]$. A block size of 1 is equivalent to the absence of the spatio-temporal block. The performance comparison results on the METR-LA dataset are shown in Figure 13a, indicating that the model performed best with a block size of 4. When the block size is set to 4, it can best capture spatio-temporal relationships without disrupting the original data flow. In comparison to the absence of the spatio-temporal block, the Test RMSE exhibited a reduction from 6.109 to 5.911, while the Test MAE demonstrated a decline from 2.985 to 2.912. Furthermore, the reduction in graph size achieved through the application of blocks resulted in an average reduction in execution time of approximately 66.9% compared to the scenario where blocks were not applied. It was demonstrated that the application of spatio-temporal blocks to the proposed model for the purpose of capturing spatio-temporal relationships is beneficial.

This improvement was observed not only in the proposed model but also when applied to STAEformer (Liu et al., 2023), a model that uses a vanilla transformer structure to capture attention. As illustrated in Figure 13b, the model also exhibited the most optimal performance when the block size was set to 4. In the case of STAEformer, the application of the block resulted in a reduction in the RMSE on the METR-LA dataset, with the value decreasing from 6.060 to 5.997. Furthermore, the MAE was also reduced, with the value decreasing from 2.970 to 2.937.

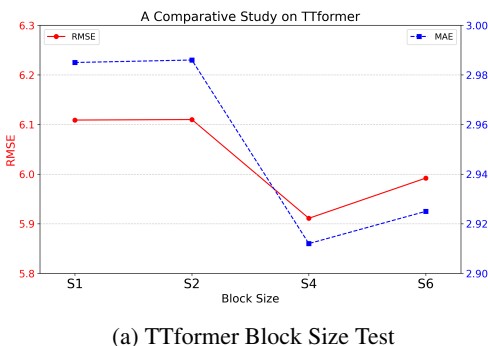

(a) TTformer Block Size Test

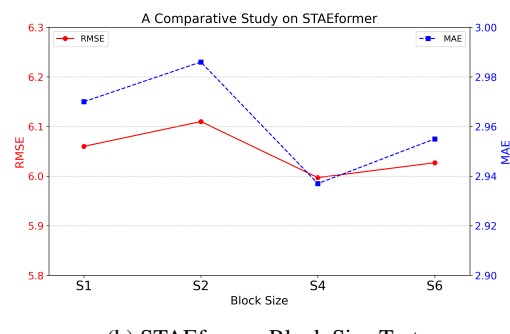

(b) STAEformer Block Size Test

Figure 13: Comparison of spatio-temporal block size

### B.3.1 STUDY OF THE SHIFTED SPATIO-TEMPORAL BLOCKS

To evaluate the performance based on the method of constructing shifted spatio-temporal blocks, we compared two approaches: shifting the given context sequence by one timestamp and shifting by one block. When shifting by one block, we provided the context sequence with a time-series of context sequence $c$ and block size $S$, constructing both regular spatio-temporal blocks and shifted

blocks. Shifting by one block results in no overlapping parts between 'blocks' and 'shifted blocks' during the calculation of the present-perfect attention module.

Comparing '1 timestamp shift blocks' and '1 block shift blocks' allows us to analyze the impact of overlapping parts on representing present-perfect relationships. When there is no overlapping, the model must learn how the context directly influences a new set of observations without explicit temporal continuity. Conversely, overlapping time series parts provide direct temporal continuity, enabling the model to understand the influence of the past on the present.

The results are shown in Figure 14, where 'w/o 1st' denotes the case when present-perfect attention, which calculates the attention between blocks and shifted blocks, is not performed. '1 block' indicates the scenario when shifted blocks are constructed by shifting one block, and '1 timestamp' represents the scenario when the context sequence is shifted by one timestamp. The actual experiment on METR-LA showed that setting the shifted block to have an overlapping part ('1 timestamp') improved performance more compared to shifting without overlapping parts ('1 block'), as illustrated in Figure 14. This suggests that overlapping time series parts provide crucial temporal continuity, enhancing the model's capability to capture the influence of past observations on present outcomes.

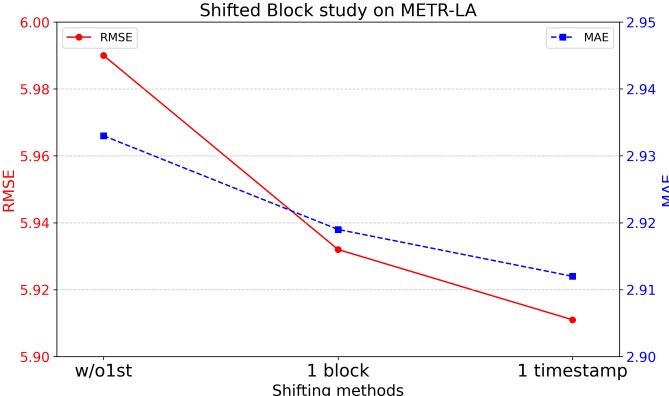

Figure 14: Comparison of shifted spatio-temporal blocks

## B.4 ANALYSIS OF CONTRASTIVE LEARNING

In the context of traffic data, sensors on the roads may record inaccurate speed values. Therefore, it is essential for prediction models to be trained to respond to outliers and contrastive learning could address this issue. To evaluate the performance of contrastive learning, we compared four versions of the model and the Figure 16 illustrates this comparison. The term 'w/o CL' refers to the scenario where contrastive learning is not applied, while 'CL_NM' denotes the results achieved by applying node masking with a node masking ratio of 0.01 and $\lambda$ set to 0.1. 'CL_NM_NF-emb' represents the outcomes when negative filtering using the spatial embedding proposed in this paper is applied in addition to node masking. 'CL_NM_NF-adj' indicates the results of conducting negative filtering using a predefined adjacency proposed in STGCL (Liu et al., 2022) after applying node masking.

In both the METR-LA and PEMS-BAY datasets, the model's performance improved when node masking was applied, compared to the model without contrastive learning ('w/o CL'). This demonstrates that the contrastive learning technique with node masking is effective in traffic data containing a significant number of zeros. Figure 15 visualizes the prediction results for specific time periods containing actual zeros in the METR-LA, comparing the 'w/o CL' and 'CL_NM' models. It was observed that the 'CL_NM' model, which underwent node masking, exhibited less sensitivity to zero values and achieved superior prediction performance.

Regarding negative filtering, while no significant performance improvement was observed for METR-LA in Figure 16a, it proved beneficial for enhancing performance in the PEMS-BAY dataset in Figure 16b. In particular, it was found that negative filtering using spatial embedding (with $nf\_threshold = 0.1$) was more effective than distinguishing hard negatives using the predefined adjacency matrix. This indicates that the similarity matrix generated through the spatial embed-

ding, learned during the joint learning process, captures more significant road correlations than the predefined road relationships based on distance.

Additionally, for the generalization test of contrastive learning, we applied the proposed contrastive learning technique to the STAEformer (Liu et al., 2023). Figure 17 presents the comparison results between the STAEformer with and without contrastive learning. These results demonstrate that the contrastive learning technique enhances the prediction performance of the STAE model in both RMSE and MAE metrics. The improved performance underscores the effectiveness of incorporating contrastive learning into spatio-temporal models like STAEformer.

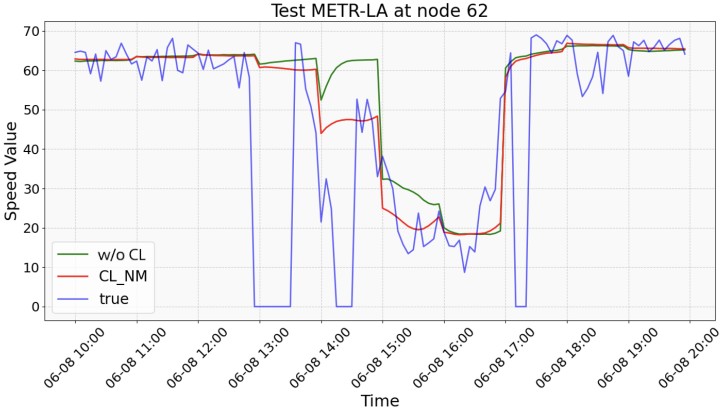

Figure 15: Evaluation of contrastive learning in traffic prediction Models

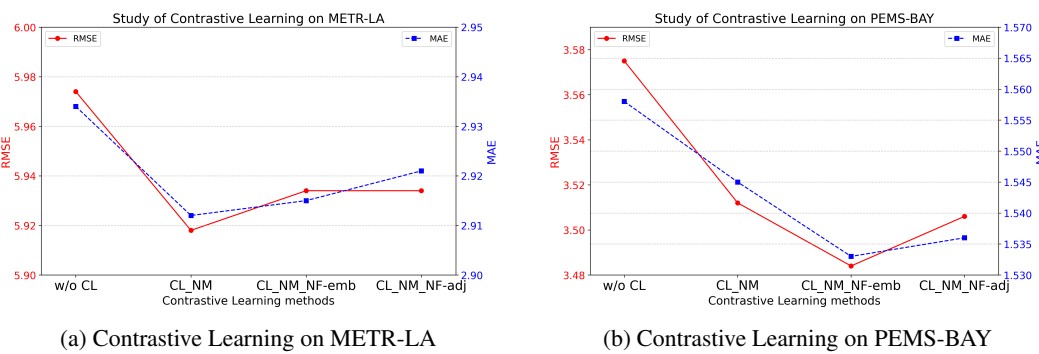

(a) Contrastive Learning on METR-LA

(b) Contrastive Learning on PEMS-BAY

Figure 16: Comparison of contrastive learning techniques

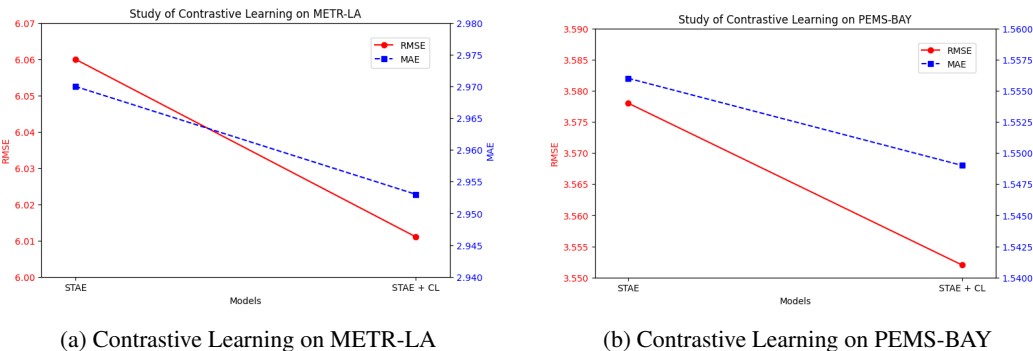

(a) Contrastive Learning on METR-LA

(b) Contrastive Learning on PEMS-BAY

Figure 17: Performance of contrastive learning for STAE