# OpenReview forum: "Tri-Tense Former: Capturing Dynamic Traffic Flow Using Tri-Tense Attention for Traffic Forecasting"
_ICLR.cc/2025/Conference — Submitted to ICLR 2025_

### Official Review · Reviewer_Rai4 · 2024-10-31

**Soundness:** 2
**Presentation:** 2
**Contribution:** 2
**Rating:** 3
**Confidence:** 4

**Summary:**

The paper proposes Tri-Tense Former, which contains three attention mechanisms—past-to-present, present, and future—that capture different aspects of traffic dynamics: the impact of historical conditions on the current state, spatial relationships between roads affecting present traffic, and the influence of current conditions on future states. Additionally, contrastive learning with predefined adjacency matrices is introduced to enhance model robustness. The model demonstrates good performance across two benchmark datasets.

**Strengths:**

1. Effective Multi-Attention Mechanisms: The network demonstrates that using distinct attention mechanisms—present-perfect and present attention—captures unique perspectives on traffic patterns, thereby enhancing forecasting accuracy.
2. Case Study Demonstration: A case study of the Dodger Stadium area highlights the effectiveness of the present-perfect and present attention mechanisms.
3. Experimental Results: The model achieves good performance on two datasets.

**Weaknesses:**

1. Key Component Descriptions: Key components, such as present-perfect attention and negative filtering, lack clear explanations. Additionally, several essential components, including the tri-tense attention module and negative filtering, are detailed only in the appendix rather than in the main paper.

2. Support for Claims: Some claims lack sufficient support. For example, it is unclear what specific inconsistencies exist in the traffic datasets and how contrastive learning addresses them.

3. Computational Cost: The three attention mechanisms likely increase computational cost, yet no runtime comparisons are provided to quantify this impact.

4. Related Work Misalignment: The related work section discusses traffic data imputation, even though the paper’s contrastive learning method is primarily intended to enhance representation quality rather than to impute missing data. If data imputation is relevant, clarifying missing data ratios in the experiments would be beneficial.

5. Terminology Issues: Certain terms could be clearer. For instance, “future attention” and “present-perfect attention” are somewhat ambiguous.

6. Clarity of Figures: Some figures, such as Figure 2, are unclear. For example, the purpose of the "?" symbols is not explained.

**Questions:**

1. How does contrastive learning contribute to more robust representations? Are there experiments to verify this robustness?

2. What specific inconsistencies are present in the traffic dataset? Providing examples would be helpful.

3. How does present-perfect attention capture the influence of past traffic conditions on the current state? How many past time steps does present-perfect attention consider? Is it limited to 𝑡−1, or does it extend further back?

4. Given that predefined adjacency matrices are typically accessible in traffic forecasting, what is the rationale for assuming they might be unavailable? And why is generating similarity matrices via spatial embeddings preferable to using a predefined adjacency matrix? Is there empirical evidence to support this?

5. Have any time comparisons been conducted to assess the computational cost of incorporating three attention mechanisms?

---

### Official Review · Reviewer_J2fi · 2024-11-01

**Soundness:** 2
**Presentation:** 2
**Contribution:** 2
**Rating:** 3
**Confidence:** 4

**Summary:**

Authors use a innovative transformer framework, Tri-Tense Former to capture spatial and temporal correlations of the traffic data. Additionally, it can handle the missing data by the contrastive learning technique. This method has a better performance than some of the baseline methods.

**Strengths:**

1. The authors use formula, figures and descriptions to explain their methodology very clearly.
2. For the ablation study, the authors clearly prove the effectiveness of the three sub components.
3. The authors use figures and tables to show the prediction accuracy of their model.

**Weaknesses:**

1. The related work section might contain s review of traffic forecasting since it is the main topic.
2. The baseline models are outdated. You might need to include more latest baseline methods (later than 2023), such as STID, PDformer.
References: Zezhi Shao, Zhao Zhang, Fei Wang, Wei Wei, and Yongjun Xu. 2022. Spatialtemporal identity: A simple yet effective baseline for multivariate time series forecasting. In Proceedings of the 31st ACM International Conference on Information & Knowledge Management. 4454–4458. In Proceedings of the AAAI Conference on Artificial Intelligence, Vol. 37. 8078–8086. Jiawei Jiang, Chengkai Han, Wayne Xin Zhao, and Jingyuan Wang. 2023. PDFormer: Propagation Delay-aware Dynamic Long-range Transformer for Traffic Flow Prediction. In AAAI. AAAI Press.
3. You need to include more dataset, such as PEMS03, PEMS04. 2 datasets are not comprehensive enough to demonstrate the great perfromance.
4. The ability to handle missing data is a key contribution. However, authors do not demonstrate how the model handle this case. Firstly, there is no missing data information of the two datasets. Secondly, the authors do not compare this method with some typical missing data imputation techniques, such as linear imputation. Thirdly, it is better to explain why missing data is a barrier for forecasting problems.

**Questions:**

1. See weaknesses.
2. Why you use 3 attentions? Could you use the traffic flow characteristics to explain your design?
3. Why you use contrastive learning for traffic forecasting? Similarly, is there any domain knowledge that motivates you to apply contrastive learning? Or, can it be treated as a general technique for spatial temporal data?

---

### Official Review · Reviewer_8qjP · 2024-11-02

**Soundness:** 3
**Presentation:** 3
**Contribution:** 2
**Rating:** 6
**Confidence:** 3

**Summary:**

This paper addresses the challenges of accurate traffic forecasting, which is crucial for intelligent transportation systems. It introduces the Tri-Tense Former (TTformer), utilizing three tense-specific attention modules to capture traffic flow across past, present, and future dimensions. To enhance model robustness against incomplete data, this paper employs contrastive learning with a negative filtering technique, independent of predefined adjacency matrices.

**Strengths:**

1.	The paper presents the Tri-Tense Former (TTformer), a novel model that uses three tense-specific attention modules to improve traffic forecasting
2.	The paper presents the concepts clearly, making the paper easy to understand. The organized format and visual elements support comprehension effectively.
3.	The paper uses real datasets (METR-LA and PEMS-BAY) for validation. Detailed ablation studies highlight the contributions of each model component.

**Weaknesses:**

1.	The writing needs to improve
a)	On page 4, line 195, the meanings of x and y conflict with the above text.
b)	In section A.6 on page 15, two paragraphs are repeated.
c)	In the experimental results, the second-best results are not underlined.
2.	The study would benefit from the inclusion of additional datasets. The reliance on only two datasets for experimentation limits the ability to fully assess the model's performance.
3.	The paper does not sufficiently compare TTformer with a wider range of existing state-of-the-art traffic forecasting models.

**Questions:**

1.	Given that each node is primarily associated with others in close geographical proximity or with similar functions, the Present Attention module and Present-Perfect Attention module calculate interactions between each node and all other nodes without utilizing a predefined or adaptive adjacency matrix. Does this approach potentially introduce excessive irrelevant information?
2.	Road 48 leads toward Dodger Stadium, while Road 16 represents outgoing routes from the stadium. However, Road 16 has a higher attention value than Road 48, which contradicts the descriptions provided in the paper.

---

### Official Review · Reviewer_4LAt · 2024-11-04

**Soundness:** 2
**Presentation:** 3
**Contribution:** 2
**Rating:** 3
**Confidence:** 5

**Summary:**

The paper proposes the Tri-Tense Former (TTformer), a novel approach designed to enhance traffic forecasting by capturing complex spatio-temporal relationships through three specialized attention modules. TTformer categorizes traffic flow into three dimensions: past-to-present (present-perfect), present, and future, each handled by its own tense-specific attention module to model dependencies accurately. The model structures traffic data into ‘spatio-temporal blocks’ with distinct key-query pairs for each module. Furthermore, contrastive learning with a negative filtering technique is employed to handle incomplete traffic data and enhance the model’s applicability across various traffic scenarios.

**Strengths:**

1. The paper is well-structured, facilitating clear understanding and readability.

2. The authors conduct comparative experiments on public, real-world datasets and provide a comprehensive analysis of the results.

**Weaknesses:**

1. The paper states that the tri-tense attention mechanism is motivated by leveraging complex temporal dependencies. However, prior works like DSTAGNN [1] and D2STGNN [2] have already addressed cross-time-slot dependencies across different nodes. Further discussion is needed to clarify the advantages of the proposed tri-tense attention compared to these existing models.

2. Comparative learning has been previously applied to traffic forecasting tasks, such as in [3] and [4]. It is necessary to discuss the specific advantages of their approach over these established methods.

3. The experimental evaluation is insufficient and should include more state-of-the-art baselines from the past three years. It is particularly important to compare the proposed model with those that utilize cross-time-slot spatio-temporal dependencies, such as DSTAGNN [1] and D2STGNN [2].

4. In the case study, it is important to compare the performance of the proposed model with state-of-the-art methods. Simply comparing predictions with the ground truth provides limited insights.

5. Using different shapes to represent nodes in a simple graph is unnecessary and may lead to confusion, suggesting that the graph is heterogeneous when it is homogeneous.

[1] Lan S, Ma Y, Huang W, et al. Dstagnn: Dynamic spatial-temporal aware graph neural network for traffic flow forecasting[C]//International conference on machine learning. PMLR, 2022: 11906-11917.

[2] Shao Z, Zhang Z, Wei W, et al. Decoupled dynamic spatial-temporal graph neural network for traffic forecasting[J]. Proceedings of the VLDB Endowment, 2022, 15(11): 2733-2746.

[3] Ji C, Xu Y, Lu Y, et al. Contrastive Learning-Based Adaptive Graph Fusion Convolution Network With Residual-Enhanced Decomposition Strategy for Traffic Flow Forecasting[J]. IEEE Internet of Things Journal, 2024.

[4] Li L, Yang K, Bi J, et al. STS-CCL: Spatial-Temporal Synchronous Contextual Contrastive Learning for Urban Traffic Forecasting[C]//ICASSP 2024-2024 IEEE International Conference on Acoustics, Speech and Signal Processing (ICASSP). IEEE, 2024: 6705-6709.

**Questions:**

Same as the weakness.

---

### Meta-Review · Area_Chair_KQsS · 2024-12-20

**Metareview:**

The authors propose a tri-tense former - a specialised attention mechanism to handle complex spatiotemporal relationships based on three dimensions -- past-to-present, present, and future.
It is unclear if the contributions of the paper are significant enough given that previous papers have proposed similar components to handle cross time-slot dependencies.
Many other ideas in the paper are also borrowed from existing works.
There was no rebuttal done.
We encourage the authors to take the suggestions on board. Please compare the proposed method with state of the art and show the advantages of the methods.

**Additional Comments On Reviewer Discussion:**

No discussion as there was no rebuttal

---

### Decision · Program_Chairs · 2025-01-22

Reject